# Transcriptomic Profiling of *Populus* Roots Challenged with *Fusarium* Reveals Differential Responsive Patterns of Invertase and Invertase Inhibitor-Like Families within Carbohydrate Metabolism

**DOI:** 10.3390/jof7020089

**Published:** 2021-01-27

**Authors:** Tao Su, Biyao Zhou, Dan Cao, Yuting Pan, Mei Hu, Mengru Zhang, Haikun Wei, Mei Han

**Affiliations:** 1Co-Innovation Center for Sustainable Forestry in Southern China, College of Biology and the Environment, Nanjing Forestry University, Nanjing 210037, China; tao.su@cos.uni-heidelberg.de (T.S.); zhoubiyao@njfu.edu.cn (B.Z.); qiuqiu121156291@outlook.com (D.C.); humei@njfu.edu.cn (M.H.); zhangmengru@njfu.edu.cn (M.Z.); yhqz212928@163.com (H.W.); 2Key Laboratory of State Forestry Administration on Subtropical Forest Biodiversity Conservation, Nanjing Forestry University, Nanjing 210037, China; 3College of Forest, Nanjing Forestry University, Nanjing 210037, China; panyuting1997@gmail.com

**Keywords:** *Fusarium*, transcriptome, defense, apoplast, invertase, sugar metabolism, *Populus*

## Abstract

*Fusarium solani* (Fs) is one of the notorious necrotrophic fungal pathogens that cause root rot and vascular wilt, accounting for the severe loss of *Populus* production worldwide. The plant–pathogen interactions have a strong molecular basis. As yet, the genomic information and transcriptomic profiling on the attempted infection of Fs remain unavailable in a woody model species, *Populus trichocarpa*. We used a full RNA-seq transcriptome to investigate the molecular interactions in the roots with a time-course infection at 0, 24, 48, and 72 h post-inoculation (hpi) of Fs. Concomitantly, the invertase and invertase inhibitor-like gene families were further analyzed, followed by the experimental evaluation of their expression patterns using quantitative PCR (qPCR) and enzyme assay. The magnitude profiles of the differentially expressed genes (DEGs) were observed at 72 hpi inoculation. Approximately 839 genes evidenced a reception and transduction of pathogen signals, a large transcriptional reprogramming, induction of hormone signaling, activation of pathogenesis-related genes, and secondary and carbohydrate metabolism changes. Among these, a total of 63 critical genes that consistently appear during the entire interactions of plant–pathogen had substantially altered transcript abundance and potentially constituted suitable candidates as resistant genes in genetic engineering. These data provide essential clues in the developing new strategies of broadening resistance to Fs through transcriptional or translational modifications of the critical responsive genes within various analyzed categories (e.g., carbohydrate metabolism) in *Populus*.

## 1. Introduction

Owing to the relatively long-life biological cycle, adaptation to various biotic stresses and environmental cues is crucial for the survival and biomass accumulation in many forest-tree species [1]. Plant disease exerts a substantial role in the environmental impacts, ecology, evolution of plants, and the efficiency of food production and bioenergy [2]. The co-evolutionary arms race between plant hosts and pathogenic microorganisms led to the development of complex molecular mechanisms for perception, defense responses, and activation against the invader [3]. Patterns of host genetic resistance, pathogen virulence, and associated favorable environments have long been postulated to collectively impact on disease outbreaks and severity in plants [4]. Therefore, the improved understanding of underlying molecular mechanisms controlling entire infection and defense between plant hosts and microbial pathogen interactions is fundamental to both science and society.

The fungal pathogenesis involves a diverse array of toxic molecules (e.g., reactive oxygen species (ROS), toxins, and oxalic acid) and many secreted proteins, including the cuticle and carbohydrate-associated degrading enzymes (e.g., cutinase and pectinase), proteases, and necrosis-inducing factors (e.g., nonribosomal peptide synthetases, polyketide synthases, and terpene synthases) [5,6]. Some of them are required for full virulence and actively release host-derived signals, termed as damage-associated molecular patterns (DAMPs), mainly including extracellular protein fragments, peptides, nucleotides, and amino acids [7]. In turn, the immune signals amplified in plant hosts are critical to promoting disease resistance after detecting fungal pathogen-associated molecular patterns (PAMPs) in the early stages of infection [8]. The pattern-recognition receptors (PRRs)-mediated perceiving of DAMPs and concurrent with additional receptor-like proteins or -kinases (RLP/Ks)-mediated signal transmission deploy the activation of host defense and immune responses, including ROS, callose deposition, mitogen-activated protein kinases (MAPKs) activation, hormone and phytoalexin, and hypersensitive response (HR) against pathogens [9]. Plants use RLP/Ks as PRRs to initiate the basal defense against pathogens upon the perception of conserved PAMPs [10]. This PAMP-triggered immunity (PTI) recognition is common to all multicellular organisms and helps the host prevent pathogen ingress quickly, efficiently, and multi-responsively, further limiting the disease spread [11]. Pathogen effectors successfully secreted to target key PTI actors can be recognized by specific intracellular disease resistance proteins evolved to trigger a basal immune response, so-called effector-triggered immunity (ETI), which is accompanied by the HR-like cell death [12]. In contrast to PTI, immune responses in ETI are faster, more robust, and prolonged than that in PTI, and both of them share almost resembling features for danger perception and defense activation [13].

The host plant perceives the apoplast-inhabiting pathogens as alien sinks and adopts means to starve pathogens in the apoplast by releasing antagonists, including hydrolytic enzymes, proteinaceous inhibitors, defense decoy, and mimicry compounds [14]. The defense responses are mounted concurrently with a profound modulation of plant metabolism, including sucrose and its split products [15]. Both host plants and pathogens maneuver over sugar supplies to acquire energy by turning the extracellular space into stiff competition. Sucrose hydrolyses involved in plant defense responses highlight a more complex modulation of sugar metabolism and signaling in pathogenesis, confirming that the defense-induced features are affected by sucrose and invertases [16]. Cell wall (extracellular/apoplastic) invertases (CWIs/CWINs) maintaining metabolizable sugars in the apoplast have been experimentally explored to be relevant to the immune response during plant–pathogen interactions [17]. Vacuolar invertases (VIs/VINs) were reported to be functionally involved in cell expansion, sugar storage, and cold-induced sweetening [18]. Non-glycosylated enzymes, cytosolic (neutral/alkaline) invertases (CIs/CINs), play multi-faceted roles in the regulation of carbohydrate metabolism, nitrogen uptake, cellulose biosynthesis, and antioxidants [19,20,21,22]. Relations between CWI expression/activity and defense response against pathogens have been extensively evaluated [23]. During the pathogen infection, modulation of VI is poorly understood due to the discordance of reports that proposed an unclear assignment of their physiological roles [16]. By contrast, very few usual studies revealed altered CIs activities in response to different pathogen infections [24,25].

Given that CWI and VI are glycosylated enzymes and intrinsically stable, their tight control may largely depend on the post-translational modification mediated by small proteinaceous inhibitors (C/VIFs, cell wall/vacuolar inhibitor of β-fructosidases). C/VIFs and the structure-related pectin methylesterase inhibitors (PMEIs) are categorized to one superfamily, enabling it with difficulties to distinguish them from sequence comparison. An increasing body of evidence implicated that post-translational mechanisms are necessary for hexoses to release to sink organs and to respond to diverse environmental stressors and phytohormone cues [26,27]. Modulation of invertase activities via C/VIF is in a spatiotemporal manner that involves multiple cellular processes, metabolic pathways, and molecular regulation [28,29,30]. Thus, CIFs are thought to be instrumental in modulating hexose leakage in the apoplast. Recent reports in *Arabidopsis* revealed that the post-translational elevation of CWI activities and its components by native inhibitors led to a marked reduction of susceptibility to bacterial and fungal pathogens [31,32,33], indicating that the fine-tuned apoplastic invertase acts as significant defense signal of the plant–pathogen interactions.

*Fusarium* species belong to a genus of filamentous fungi that consist of agronomically important plant pathogens, and many of them are mycotoxin producers [34]. Of these pathogens, *Fusarium solani* (Fs) is a soil-borne root-infecting fungal pathogen that invades the plants through roots and colonizes the xylem vessels, leading to malfunction of upward water and nutrients translocation. *Fusarium* is the primary causal agent of diseases in root necrosis, wilts, and cankers, leading to massive yield and quality losses in a broad range of crops and forest organisms, including perennial *Populus* [35,36,37]. In the *Fusarium* genome, due to unspecific distributions of secondary metabolism-related biosynthetic gene clusters and pleiotropic effects of individual genes with minor contributions to the pathogenicity, assigning definitive roles in pathogenesis and virulence remains a challenging task [38]. Therefore, manipulating the host resistance is the primary strategy to reduce *Fusarium* pathogen’s infection and damage. Although the defense response against Fs has been explored in many crops, little is known about the profiling of root defense response in susceptible *Populus trichocarpa*, a woody model plant. Additionally, expression patterns of invertases and the coordinated C/VIFs in shaping apoplast dynamics remain unknown. In this study, transcriptomic sequencing was used to analyze the dynamic changes of gene transcripts in *Populus* roots at 0, 24, 48, and 72 h post-inoculation (hpi) of Fs infection. We aimed to characterize highly differentially expressed genes (DEGs) during the host–pathogen interaction to reinforce the understanding of underlying defense mechanisms in *P. trichocarpa*. As primary sugar metabolism-related enzymes, expression patterns of invertase and invertase inhibitor-like families were further evaluated. We pointed out a peak of DEG at 72 hpi in the Fs-challenged *Populus* roots, highlighting the importance of the extracellular hexose retrieval for the interaction outcome. Our study provides insight into systematic defense adaptation regulations through signal transduction, pathogenesis, secondary metabolite biosynthesis, and carbohydrate metabolism in woody plants.

## 2. Materials and Methods

### 2.1. Plant Materials and Fungal Inoculation

*P. trichocarpa* (genotype *Nisqually-1*) was cultured under in vitro conditions (25 °C, 16/8 h day/night photoperiod, 20 µE) on standard McCown^T^ woody plant medium (WPM) (Duchefa, Haarlem, The Netherlands) with 30 g L^−1^ sucrose, 0.1 mg L^−1^ IBA, and solidified with 8 g L^−1^ plant agar. For the fungal pathogen infection, an inoculum of *F. solani* was grown for seven days on potato dextrose agar (PDA) at 25 °C with a 12 h photoperiod [39]. Conidia were collected by rinsing plates with sterile water, scraping the agar surface with a scalpel, and filtering the conidial suspension through the sterile micro cloth. Roots peripheral areas of 8 week in vitro cultured plants were inoculated with 20 µL *F. solani* spore suspensions (1.0 × 10^6^ spores/mL) by pipette for 0, 24, 48, and 72 h using a chamber.

### 2.2. Fungal Growth, DNA Isolation, and Quantification

The conidia growing at standard WPM medium for 0 h, 24 h, 48 h, and 72 h was collected and photographed on the Zeiss M2 microscopy (Zeiss, Aalen, Germany). According to the manufacturer’s instruction, DNA from *Populus* roots inoculated by Fs was extracted using TIANamp Yeast DNA Kit (Tiangen, Beijing, China). The quality and quantity of DNA were evaluated by measuring the concentration (ng/μL) and A260/A280 and A260/A230 via a NanoDrop 2000 spectrophotometer (Thermo, Shanghai, China). The genomic DNA was subsequently used as the template for the fungal DNA quantification using quantitative real-time PCR (qPCR). DNA quantities are reported as ng of fungal DNA obtained from root tissues. According to the instrument technical manual, the determination was quantified by the Step One Plus Real-Time PCR System (AB, Foster City, CA, USA) based on the linear regression equation generated from serial dilutions (1:1, 1:50, 1:100, 1:400, and 1:800) with *R*^2^ = 0.98. Fungal DNA (20 ng) deriving from Fs was serially diluted in sterile water, and 20 ng of each root DNA sample was compared to the dilution standard curve to determine fungal DNA quantity.

### 2.3. RNA Isolation and Library Preparation

Total RNA of Fs infected root tissue was extracted using the TRIzol reagent (Thermo Fisher, Waltham, USA) following the manufacturer’s protocol. RNA integrity was evaluated using a NanoDrop 2000 spectrophotometer (Thermo, Shanghai, China). All sequenced samples were generated from high-quality RNA samples with a 28S/18S ≥ 1 and an A260/A280 between 1.8 and 2.1. Strand-specific RNA-Seq libraries were prepared using an Illumina Standard RNA Sample Library Preparation Kit (Illumina, San Diego, CA, USA) according to the manufacturer’s instructions. The deep cDNA sequencing was performed on an Illumina Hi-Seq 2500 platform (Oebiotech, Shanghai, China). A total of 12 independent libraries were constructed for each time point of Fs infection. These libraries were then sequenced on the Illumina sequencing platform (HiSeq 2500 or Illumina HiSeq X Ten), and 125–150 bp paired-end reads were generated.

### 2.4. Transcriptomic Sequencing, Data Mining, and Bioinformatics Analyses

Clean reads were mapped to the *Populus trichocarpa* v3.1 (Poplar) genome (https://phytozome-next.jgi.doe.gov/info/Ptrichocarpa_v3_1). The different values of each gene were calculated using Cufflinks, and the read counts of each gene were generated by HTSeq-count (0.11.11, 03/01/2019) [40,41]. The fragments per kb per million reads (FPKM) of each gene was calculated using HISAT2 software (2.2.1, 7/24/2020) [42]. Differential expression analysis was performed using the DESeq2 Bioconductor package R (3.12, 28/10/2020) [43]. Only genes with FPKM > 1 were used for the differential expression analysis, and the significance of differences in expressed genes was judged on the *p* < 0.05 and fold change of normalized base mean value (FC) > 2 or FC < 0.5 that was set as the threshold for significantly differential expression. Hierarchical cluster analysis of differentially expressed genes (DEGs) was performed to demonstrate gene expression patterns in different groups and samples. The Gene Oncology (GO) enrichment (10.5281/zenodo.2529950, 01/01/2019) determined the functional categories using Fisher’s exact test [44], and the Kyoto Encyclopedia of Genes and Genomes (KEGG) pathway (http://www.genome.jp/kegg/) enrichment (96.0, 01/10/2020) analysis of DEGs was performed, respectively, using R based on the hypergeometric distribution [45]. Both Fisher’s exact test (*p* < 0.05) and multi-test adjustment (false discovery rate (FDR) < 0.05) were applied.

### 2.5. Evolutionary Analyses and Expression Evaluation by qPCR

The phylogenetic tree was constructed by MEGA X (https://www.megasoftware.net/) (10.2.2, 01/10/2020) [46]. The evolutionary distances were computed using the Poisson correction method and are in the units of the number of amino acid substitutions per site. The percentage of replicate trees in which the associated taxa clustered together in the 1000 bootstrap test is shown next to the branches. The phylogenetic analyses involved 100 full-length protein sequences, of which 70 were from *Populus*, and 30 from *Arabidopsis*. By inputting the gene name, accession numbers are available from the Arabidopsis Information Resource database (TAIR, https://www.arabidopsis.org/). Total RNA of *Populus* roots was extracted using the RNeasy Plant Mini Kit (Tiangen, Beijing, China) from respective tissues according to the manufacturer’s instructions. RNase-free DNase I (Tiangen, Beijing, China) was used to remove genomic DNA. First-strand cDNA was synthesized using the PrimeScript II 1st Strand cDNA Synthesis Kit (Takara, Beijing, China). For a standard qPCR technical application, the samples were loaded to a TB Green Premix ExTap (Tli RNaseH Plus) (TaKaRa, Osaka, Japan). The PCR reactions were run on Step One Plus Real-Time PCR System (AB, Foster City, CA, USA) with a three-step PCR procedure, according to the manufacturer’s instructions. Primers were evaluated with dilutions of cDNA, producing an *R*^2^ value ≥ 0.99. The relative expression level of a target gene was calculated by normalizing to the geometric mean (geNorm) of multiple reference genes, and values of the control (0 hpi) were set up at 1 [47]. The *PtUBIC* (Potri.006G205700), *Ptβ-Actin* (Potri.019G006700), and *PtEF1-α* (Potri.006G130900) were used as the reference genes. Each experiment was performed with three biological repeats, and each sample was conducted with three technical replicates. The heatmaps were constructed using the online CIMminer program (http://discover.nci.nih.gov/cimminer/home.do) (19/07/2018). Primer sequences used for gene expression analyses are listed in Appendix A.

### 2.6. Invertase Extraction and Determination of Invertase Activities In Vitro

The in vitro determination of invertase activities was performed according to the previous report [29]. Selected *Populus* root tissues were ground in liquid nitrogen and homogenized in 500 μL extraction buffer, pH = 6.0 (alternatively, pH = 7.5 for CI isolation). After centrifugation, the supernatant was collected, and the pellets were washed once with extraction buffer and twice with extraction buffer only. The cell wall pellets were re-suspended in 500 μL assay buffer (pH = 4.6) and used for the determination of CWI activity. Endogenous sucrose in the supernatant (VI) was removed by cold acetone precipitation (20 min, −20 °C). After centrifugation, the pellets were resuspended and dissolved in assay buffer. For the measurement of enzyme activity, 100 μL of the obtained preparations was incubated with 100 μL sucrose (100 mM) and deionized water up to 300 μL. After incubation for 1 h at 37 °C, the reaction was terminated by the addition of 30 μL sodium phosphate (1 M, pH = 7.5) and heating at 95 °C for 5 min. The assay was performed in quadruplicate, one of which was neutralized and boiled immediately after sucrose addition. Free glucose was measured in a coupled enzymatic–optical assay. According to the Lambert–Beer law, the formation of NADPH was measured spectrophotometrically at 340 nm, and the liberated glucose was calculated. Invertase activity was expressed in n kat g^−1^ fresh weight (1 nkat = 1 nmole glucose liberated per second).

## 3. Results

### 3.1. Quantification of the Fungal Growth and Colonization in Populus Roots

The Fs growth was initially monitored using microscopy within 4 days after inoculation in WPM. The Fs produced fewer conidia in the culture medium at 24 hpi than that at 48 and 72 hpi, which might contribute to the relatively lower levels of infection and fungal DNA in roots over a different time course. After 24 hpi, the germinated conidia began to prolong and disperse widely (Figure 1a). The specific Fs translation elongation factor-1 alpha (*EF-1α*) gene was used to inspect fungal DNA abundance by qPCR as an indicator of the Fs infection progression roots. In contrast to 0 hpi, the quantity of *EF-1α* DNA in pathogenic inoculated samples revealed significant differences between the times of sampling of 48 hpi and 72 hpi (Figure 1b). The highest levels of fungal DNA were detected at 72 hpi for the interaction with the Fs, suggesting the established colonization of the fungal pathogen.

Furthermore, wilt progression in *Populus* roots infected with Fs to different degrees was examined by phenotypic screening at 0, 24, 48, and 72 hpi (Figure 1c). The minor symptoms occurred at 24 hpi in plants inoculated with Fs, as quantified by qPCR analyses (Figure 1b). After 24 hpi, the necrosis symptoms in roots infection started to appear, and the disease incidence reached a scale of all eight plants, confirming the establishment and colonization of the fungal disease in vascular tissues of the roots. After 72 hpi, all treated *Populus* plants showed visible necrosis in the roots, whereas the shoots appeared to be insignificantly affected and remained healthy and robust (Figure 1c). These profiles indicate an enhanced reaction of compatibility between the host and the pathogen at the later stages of inoculation, suggesting that the plant may already initiate the molecular response to the pathogenic invader at 24 hpi (Figure 1a). When the infection was established, the host plant defense responses and metabolite changes at the early-intermediate stages and the detection of the modified transcripts in the pathogen were both considered to be necessary.

### 3.2. Global Changes of Gene Expression in the Host Transcriptome

By quantifying the Fs colonization, the comprehensive gene transcript profiles of *Populus* roots and concurrent with its pathogen at the different stages of infection (0, 24, 48, and 72 hpi) were analyzed via a high-throughput RNA sequencing technique. Analyses of the fungal DNA showed a constant increase during the infection, as evaluated by qPCR (Figure 1b). However, the mapping of fungal RNA-seq to the Fs reference genome was not successful owing to the limited number of aligned read counts, indicating that the fungal DNA quantities were detected particularly at the later stage of the inoculation (i.e., 72 hpi). Therefore, downstream transcriptomic analyses and visualization were only performed for the host *Populus*.

The transcriptomic sequencing of Fs infected *Populus* roots were conducted using the Illumina Hi-Seq 2500 high-throughput sequencing platform, showing an average of 48.65 million (M) raw reads, ranging values from 42.16 to 58.71 M per library obtained from the four treatments (FS0, FS24, FS48, and FS72), including three biological replicates (a, b, and c) for each condition (Table 1). After removal of the low-quality reads, adapter sequences, and duplications, a total of 81.26 gigabytes (G) clean bases was obtained to reach an average value of 6.8 G, ranging from 5.86 to 8.15 G, and the percentage value of Q30 base was 93% or more. Based on the FPKM values, variabilities of the density and composition of data sets were compared statistically, showing that FPKM values between 0.5 and 1 had the largest proportion, followed by FPKM values over ten within each sample (Figure 2a,b). Using principal component analysis (PCA), the transcriptomic background and distribution patterns were evaluated according to normalized read counts. The spot indicates good reproducibility of transcriptome data within the same condition. As depicted in Figure 1c, FS72 shows a substantial distance from the other three treatments based on the dimension of principal component 1 (PC1) (91.28% variance). Combined with the dimension of PC2 (5.96 variance), the PCA confirmed the significant variation of the distribution and reliability of the global gene expression data. The correlation values of FPKM between each pair of samples were evaluated by Pearson’s correlation tests, ranging values from 0.91 to 1.00 (Figure 2d). Overall, these results reveal a high correlation between different treatments and biological replicates, allowing us to extensively mine the global gene expression profiling via the transcriptomic sequencing data.

### 3.3. Differential Expression and Functional Classification of Fs-Responsive Genes

The clean data of transcriptomic sequencing of Fs infected roots at different time courses were assembled according to the *P.trichocarpa* reference genome (v3.1) set using HISAT2 software. A total of 41,335 unigenes was successfully mapped, showing the total mean reads lengths of 41,611,336 bp (96.88%), 47,291,286 bp (95.41), 33,349,868 bp (66.96%), and 30,957,885 bp (63.10%) for FS0, FS24, FS48, and FS72, respectively (Appendix A). Using the false discovery rate (FDR) calculation, the FC more than 2 (Log_2_FC > 1), *p* < 0.05, were adjusted as the threshold for evaluation of DEGs. Thus, the expression modes of all DEGs between six compared groups were revealed (Table 2). Approximately 1659 genes exhibited significant changes, including 979 genes of upregulation and 680 genes of downregulation at the early stage of infection (FS24/FS0). After 48 hpi (FS48/FS0), the total number of DEGs increased two-fold (3479), of which 2669 and 810 genes were significantly induced and repressed, respectively. In contrast, a total of 4296 DEGs, comprising of 2870 upregulated and 1426 suppressed genes, were significantly affected at 72 hpi (FS72/FS0), suggesting that many more genes highly responded in the later stage of Fs infection. Most of the DEGs with significance were distributed at this time point, ranging the Log_2_FC from 1 to 5 (Figure 3a). Analyses in the heat map illustrated the association of clustered DEGs (upregulation/downregulation) between 0 hpi and 72 hpi (Figure 3b).

DEGs’ mining revealed that the transcripts of 552 genes were significantly altered within the three different comparisons (FS24/FS0, FS48/ FS0, and FS72/ FS0) (Figure 3c). A total of 63 genes were identified to represent the most highly responsive genes at all six comparisons (Table 2), of which three genes (e.g., expansin-A11, L-ascorbate oxidase, and glucan endo-1,3-β-glucosidase) were markedly suppressed (Figure 3d, Appendix A). Among these highly promoted genes, approximately 20 unique genes exhibited drastic increases (*p* < 0.01, Log_2_FC ≥ 5), including some transcription regulators (e.g., WRKY, F-box protein, U-box protein, and MYB), signaling proteins, transporters, carbohydrate and lipid metabolism (e.g., β-glucosidase, mannitol dehydrogenase, UTP-glucose glucosyltransferase, and phospholipase), oxidation-reduction (e.g., cytochrome P450, glutathione S-transferase, and zinc-binding dehydrogenase), and the secondary metabolism-related genes. Interestingly, the number of DEGs increased to 238 at all six comparisons when the parameters of FC were set up as 1.5 (*p* < 0.01) within all compared groups (Appendix A).

To understand the biological processes (BP) and molecular function (MF) classification associated with host *Populus* reactions to Fs infection, the GO enrichment (log_2_FC > 1, FDR < 0.05) was performed to determine the functional categories. A total of 19,951 DEGs were assigned to the different GO annotations and were further categorized into 61(FS24/FS0), 79(FS48/FS0), 90(FS72/FS0), 101(FS48/FS24), 91(FS72/FS24), and 57(FS72/FS48) functional GO classification, containing respective gene numbers of promotion and repression (Table 3 and Appendix A). These annotated GO terms were majorly categorized into BP, cellular component (CC), and MF. At the later stage of Fs infection (FS72/FS0), the largest proportion of 2275 DEGs were classified into the cellular process (754) and metabolic process (1263), biological regulation (253), catalytic activity (1217), binding (1147), and single-organism process (Figure 4a). Moreover, genes associated with oxidation-reduction (GO: 0055114), protein phosphorylation (GO: 0006468), and regulation of transcription (GO: 0006355) in the category of BP were the top three among significantly enriched GO terms. Genes related to the membrane (GO: 0016020), membrane integral component (GO: 0016021), and cell wall (GO: 0005618) were the most abundant GO term in the CC category. Among the category of MF, genes encoded protein kinase (GO: 0004672), hydrolase (GO: 0004553), and transcription factor (TF) (GO: 0003700) showed the predominant patterns in response to Fs infection (Appendix A).

Using the KEGG pathway database, DEGs involved in the critical intracellular metabolic processes and biological pathways between the host and fungal pathogen interaction were further mapped. Under the defined condition (FDR < 0.05), a total of 200, 531, and 566 DEGs for FS24/FS0, FS48/FS0, and FS72/FS0, respectively, were categorized into 20 top KEGG terms (Figure 4c,d). These top KEGG terms were primarily comprised of carbohydrate metabolism (e.g., peu00500, peu00520, and peu00040), biosynthesis of secondary metabolites (e.g., peu00940, peu00902), signaling transduction (e.g., peu04016, peu04626), and amino acid biosynthesis (e.g., peu00270, peu00300, and peu00350) (Table 4). Similarly, the KEGG analyses revealed that many DEGs exhibited more significant changes in expression levels, particularly at 72 hpi compared to other time courses of Fs inoculations.

### 3.4. Expression Validation of Invertase and Invertase Inhibitor-Like Families and Enzyme Activities

Given findings that the carbohydrate metabolism was the most predominant pathway during the different time courses of Fs infection, invertase and the coordinated invertase inhibitor families were chosen to evaluate their expressions via quantitative real-time PCR (qPCR). As a representative sucrose hydrolysis-related enzyme, six extracellular CWIs, three VIs, and twelve CI-encoding genes were characterized in the *P.trichocarpa* genome [48]. Based on the functional homologs in *Arabidopsis* and crops, a total of 49 *Populus* invertase inhibitor-like genes (C/VIFs) have been mapped, including two recently characterized inhibitors and some PMEI genes [39]. Using homologous isoforms in *Populus* and *Arabidopsis*, the phylogenetic analyses revealed a more conserved feature between three types of invertase than C/VIFs and PMEIs, showing divergent branches (Figure 5a). Among the DEGs (FPKM > 5) depicted in a heatmap (Figure 5b), approximately 29 genes, including three CWIs (*PtCWI3*, *4*, and *5*), two VIs (*PtVI2* and *3*), seven CI/CINVs (*PtCI1*, *2*, *3*, *5*, *8*, *9*, and *12*), and fifteen C/VIFs and PMEIs, exhibited significantly altered values during the four time courses of Fs infection. The validated results of gene expressions by qPCR were mostly compatible with the transcriptomic sequencing data, confirming 18 DEGs (eight upregulated and ten downregulated) upon Fs inoculation (Figure 5c). Most genes in the PMEI family showed continuously increased expressions during the four examined time points upon Fs infection. Further analyses of extracted enzyme activities in *Populus* roots showed that CWI increased more markedly at 48 hpi (>3 folds) and 72 hpi (>4 folds) than 0 hpi (Figure 5d). In contrast, activities of VI and CI appeared to be moderately affected. The slightly elevated CI activities might result from the increasing of their encoding genes transcripts. Interestingly, enzyme activities were discordant with their transcripts for CWI and VI, prompting that the modulation of CWI and VI activities may depend much on post-translational control due to the downregulation of invertase inhibitors (*PtC/VIF1*, *2* and Potri.007G108300) observed at the same time points. Nevertheless, it is possibly that both host and pathogen manipulated the inhibitory potential of these invertase inhibitors, leading to fine-tuned activities of the sugar-splitting enzymes in their favor.

## 4. Discussion

Most fusaria are generally categorized as hemibiotrophic pathogens as the infection is initiated by an intermediate biotrophic lifestyle and eventually enters a necrotrophic mode to kill host cells using toxic compounds, enzymes, or microRNAs [49]. They employ a broad spectrum of infection strategies to release destructive *Fusarium* diseases and cause root rot and sudden death in some trees [50,51]. *P. trichocarpa* is a perennial woody model plant. However, no studies had been attempted to understand the mechanisms and pathways involved in response to *Fusarium* pathogen attack. The transcriptomic scale responses in *Populus* roots against Fs infection also remain unknown, making it challenging to propose a pathogenic molecular analysis and unravel the biological roles of a specific gene(s). Our present work provided the first large-scale survey of global gene transcripts changes when *Populus* roots were challenged with Fs.

Based on the transcriptomic analyses, a total of 923 *Populus* genes were identified to be highly differentially expressed in response to the Fs infection (FS72/FS48), and they encode for enzymes that have been reported to be commonly involved in defense-related networks. Of these genes, approximately 839 were drastically enriched in 29 GO terms, and 240 were categorized into 21 KEGG pathways (Appendix A). We placed these DEGs into five broad classifications, including the signaling transduction, defense responsive-related, transcriptional regulation, biosynthesis of secondary metabolite, and carbohydrate metabolism enriched pathways. We further explore their transcript abundance and discuss their putative molecular functions below. Although the other DEGs were enriched into defined categories or bins are not considered here, their known relevance in the defense response literature remains worth further investigation.

### 4.1. Changes in Signaling Transduction Genes

Plant hosts recognize potential pathogen surface-derived molecules through sensors, known as PRRs, that initiate a serious defense response and relay the signal downstream to convergent signaling pathways, triggering broad-spectrum immunity [10,11]. Eukaryotic protein kinases (PKs) are a superfamily that facilitates this signal transduction by catalyzing phosphate transfer to free hydroxyl groups of serine/threonine/tyrosine residues substrate [52]. These PKs are represented by several families belonging to groups of RLKs, MAPKs, and calcium-dependent protein kinases (CDPKs) [53]. Based on signature motifs in the ectodomains, transmembrane RLKs can be categorized into more than 12 groups in *Arabidopsis*. The leucine-rich repeat RLKs (LRR-RLKs) are the largest family that play crucial roles in growth and development and stress-responsive signals in plants [54]. Some representative types are comprised of the G/S- locus lectin receptor-like kinases (LecRLKs), wall-associated kinases (WAKs), extensin-like, self-incompatibility domain (S-domain), cysteine-rich repeat secreted proteins (CRRSPs), thaumatin-like proteins (TLPs), leaf rust kinase-likes (LRKs), and other unknown receptor kinases. A few of them exerting essential roles in plant innate immunity have been unraveled [55]. A total of 379 LRR-RLKs and 231 LecRLKs genes have been identified in the *Populus* genome based on the genome-wide survey [56,57]. Our study identified 141 various RLKs that showed differential expression patterns in *Populus* roots infected by Fs (FS72/FS48). Among these DEGs, two LRR-RLKs-encoding genes (Potri.005G043700 and Potri.016G011400) and a putative LecRLK (Potri.T021500) were significantly suppressed. These findings were compatible with a recent report that revealed similar expression patterns of LRR-RLKs in response to soli endophytic fungi [58]. Three homologs of *Arabidopsis* WAK2 showed marked changes, suggesting that they appeared to be promoted particularly upon the pathogenic fungi/bacteria in trees [59]. Along with two significantly suppressed TLPs genes (Potri.012G047800 and Potri.017G075500), specific inductions of five CRRSP-encoding genes were identified, two (Potri.007G120300 and 600) of them downregulated after Fs inoculation at the same time points.

Pathogen recognition triggers the activation of the host defense mediated by MAPK signaling cascades reported to regulate numerous physiological processes, including downstream signaling specificity, innate immunity, hormone, abiotic stress, and pathogenic defense responses [60]. In line with the widely accepted model, we found five MAPK/KK/KKK-encoding genes induced in response to the Fs at 48 hpi, and this number was expanded to six, owing to an additional one (Potri.009G066100, *MAPK3*) promoted at 72 hpi. A recent report revealed that *MAPK3* might involve the negative regulation of callus formation in *Populus* roots [61]. The sucrose non-fermenting (SNF1) protein kinases regulate carbohydrate metabolism, providing crosstalk between the hormone and sugar signaling pathways [62]. Additionally, two putative SNF1 genes were identified to be highly expressed at 72 hpi. Furthermore, the functional roles of phytohormone signaling pathways are well established to deploy the systematic acquired resistance.

Emerging evidence suggests that resistant mechanisms of plant hosts to pathogenic *Fusarium* is mediated by phytohormone signal transduction pathways, including salicylic acid (SA), abscisic acid (ABA), jasmonate (JA), ethylene (ET), and auxin [63,64]. A total of 26 DEGs were involved in phytohormone signaling in *Populus* roots at 72 hpi compared with 48 hpi. It was well-known that *Fusarium oxysporum* infection activated the transcription of auxin-related genes prompting a higher auxin biosynthesis [65]. Some of them containing four indole-3-acetic acid-amido synthetase GH3.1 genes, an auxin-induced gene (Potri.001G177500), and an auxin-responsive gene (Potri.018G127800, *IAA29.2*) were significantly depressed. The latter one was in line with a more recent report that revealed a decreased expression of *IAA29.2* during nitrogen remobilization [66]. Other enriched genes showing specific increases in the phytohormone signaling category contain two protein phosphatase 2C (PP2C) and three ET responsive transcription factors (ERFs). In plants, biochemical and molecular genetic studies in various species have identified PP2C enzymes as critical players in ABA signal transduction. ERFs play crucial actions in regulating adapted environmental stresses, conferring enhanced disease resistance against fungal pathogens by modifying their expressions [67]. A higher number of them were significantly induced at 48 hpi (55) and 72 hpi (41) than that in 24 hpi (10), suggesting that ERFs might be activated mainly in the later infection stages of Fs. Among the core JA signaling-related genes, three JA zim-domain-containing proteins were strongly expressed at the same time point. These results are supported by the findings that JA signaling genes were functionally characterized in poplar infected with leaf rust pathogen [68]. Two additional homologs, a two-component response regulator and a xyloglucan endotransglucosylase/hydrolase, were identified to be substantially promoted and depressed, respectively.

### 4.2. Expression of Defense Responsive and Transcriptional Regulation-Related Genes

The typical plant defense responsive features involve pathogenesis-related (PR) proteins and antimicrobial peptides. Despite the advance of new scientific tools, PR proteins and peptides have been isolated much before; however, their biological role remains largely enigmatic. PRs were classified into 17 families that have been widely explored to unravel their stress tolerance function and fungi toxic [69]. In contrast to 48 hpi, a high number of genes representing different PRs—including two homologs of hevein proteins (PR4 family), two proteinase inhibitors (PR6 family), four thaumatin-like proteins (PR5 family), five lignin-forming anionic peroxidases (PR9 family), twelve chitinases (PR3 family), two lipid transfer proteins (PR 14 family), and one defensin-like protein (PR12 family)—were identified as up- and downregulated significantly after inoculation at 72 hpi. In plants, germin and germin-like proteins constitute diverse family proteins. Some of them in defense against pathogen attack and stress response had been proposed based on gene functional and genetic approaches. Six of 25 germin-like proteins were observed to have the same trend of drastic increase in response to Fs inoculation, suggesting their potential roles in defense response in *Populus*. Additionally, 11 ankyrin repeat family-related proteins and one homologous protein containing a calcium-dependent phospholipid-binding domain were also strongly induced at 72 hpi. The snakins are the foremost cysteine-rich defense and antimicrobial peptides in plant hosts [70]. Three homologs of *Snakin-1* showed significantly decreased expressions. Overall, the profiling of signaling and defense-related genes indicates that specific gene expressions were more pronounced toward the Fs infection. The induction and accumulation of PR, cell wall-degrading enzymes, and proteinase inhibitor genes suggested that the degradation of cell wall components of pathogens and proteolysis inhibition are important defense reactions in *Populus* roots against *Fs* at the specific infection stages (48 hpi and 72 hpi).

Transcriptomic analysis identified a large number (177) of TFs-encoding gene families—including MYB, WRKY, NAM-ATAFI/2-CUC2 (NAC), basic helix-loop-helix (bHLH), basic leucine zipper (bZIP), GATA, and ERFs—found that their expression was significantly changed in response to Fs inoculation at 72 hpi. The MYB family protein-encoding genes known as TFs, harboring MYB domains, can be classified into four types (4R-MYB, 3R-MYB, R2R3-MYB, and MYB-related). An *Arabidopsis* R2R3-MYB transcription factor, AtMYB30, functions as a positive regulator of HR-like cell death and pathogen invasion [71]. Recently, 207 putative R2R3-MYB genes were mapped in the *Populus* genome based on the repeated sequences and conserved domains. *Arabidopsis* R2R3-MYB TF directly targets the flavonoid biosynthesis genes and the signaling chain, leading to the activation of flavonol gene transcripts in the phenylpropanoid biosynthesis pathway [72]. In line with this, we also observed differential expression of genes encoding enzymes related to the biosynthesis of flavonol and phenylpropanoid (see below). In our study, MYB represented the largest number of various TFs families. Seventeen MYB-encoding genes were highly expressed upon Fs infection at 72 hpi compared to 48 hpi, eight downregulated. Twelve bHLH and eight bZIP-encoding genes were identified, showing significant changes. A more recent rice study revealed that a bHLH transcription activator modulated defense-related signaling [73]. Other strongly represented TF families were NAC TFs. Along with six WRKY and five GATA- encoding genes, eleven NAC TFs exhibited significant induction. Following our results, a *NAC13* (Potri.001G404100) was recently characterized to confer salt tolerance in *Populus* through a molecular assay, suggesting the roles in stress adaptation. Due to dual functional roles, *Arabidopsis* NAC facilitates regulation of both JA- and ABA-dependent responses and manipulates plant stress responses by activating other genes coding for MYB TFs, amylase, cold-responsive proteins, dehydration responsive proteins, glutathione-S-transferases, and late embryogenesis abundant (LEA) proteins [74]. ERFs are well-known to be involved in ethylene signaling. The 1-aminocyclopropane-1-carboxylate oxidases (ACO), participating in the last step of ET biosynthesis, was associated with resistance against wilting caused by vascular diseases [75]. In agreement with identifying several ERFs (see above), the Fs infection upregulated a 1-aminocyclopropane-1-carboxylate oxidases (ACO) gene and suppressed two ACO genes upon inoculation at 72 hpi. The induction of ACC oxidases was revealed only by the pathogenic *Fusarium*, indicating that ET might be directly involved in signaling the host defense control.

### 4.3. Genes Involved in Modulation of Secondary and Sugar Metabolism

As previously stated, the biosynthesis of secondary metabolite and carbohydrate metabolism remain predominant pathways containing the largest number of DEGs within the top KEGG classification at 72 hpi (Figure 4). Transcriptomic results illustrated that the transcriptional modulation of multiple targeting genes involved both in the upstream phenylpropanoid pathway and downstream flavonoid and diterpenoid biosynthesis pathways. The apoplast is the first line of defense for plant hosts, defining the primary strength to restrict pathogen penetration to the cell. Pathogens attacking the host cell wall trigger the biosynthesis of phenylpropanoid pathway compounds, which range from inducible physical/chemical barriers against infection to the signaling molecules involved in signaling induction of the defense gene in hosts [76]. The entire metabolomic pathway is a complex network regulated by multiple gene families. In our study, transcriptomic data revealed that many phenylpropanoid-related genes showed significant changes at 72 hpi compared with 48 hpi. More specifically, these genes comprised homologs of a cinnamoyl-CoA reductase (CCR), a cinnamyl alcohol dehydrogenase (CAD), a caffeoyl 3-o-methyltransferase (COMT), a cinnamate 4-hydroxylase (C4H), nine berberine bridge enzyme-like (BBE) enzymes, 10 β-glucosidases, and 15 peroxidases. More recently, the same *Populus* C4H gene (Potri.018G146100) was found to be upregulated in gall formation by root-knot nematodes, suggesting the potential role of lignification and defense regulation in vascular tissues [77]. The transcriptomic data also revealed 14 and 5 markedly modified genes related to the flavonoid and iso-flavone biosynthesis pathway, respectively, upon the Fs inoculation. Among these genes, eight of them belonging to flavonoid 3′-monooxygenases, along with two UDP-glucose flavonoid 3-O-glucosyltransferases (UFGT), one flavonoid 3- hydroxylase (F3H)/flavonol synthase (FS), as well as four terpene synthases (TPs) that were identified. Terpenoids are a large and structurally diverse class of metabolite molecules and have been shown to exert significant defensive and developmental roles in numerous plant species [78]. These findings further supported the notion that active regulation of the cell wall strengthening and lignification and synthesis of secondary metabolites (e.g., terpenoids, flavonoids, and iso-flavonoids) facilitates the core structure of host resistant mechanisms between the *Populus* and pathogen interaction in other pathological systems [76,79].

Pathogen infection is the first impetus to activate defense responses concurrently by a rapid induction of primary metabolism in sink organs, either satisfying the increased capacity for carbohydrates or maintaining the cascade of natural defense response-mediated stress adaptations through surveillance of cellular intactness. In turn, pathogens try to manipulate plant carbohydrate metabolism for their own needs as energy sources [80]. The intense nutrient competition occurs at the plant–pathogen interface, in which activities of sugar metabolism-related enzymes and transporters from plant hosts or/and pathogen appear to be critical for the outcome of the interaction [81]. In our study, except for ten genes (β-glucosidases) in the crosstalk of the phenylpropanoid pathway, a total of 46 significantly expressed genes were identified based on the GO term of the carbohydrate metabolic process and pathways related to sugar metabolism from 48 hpi to 72 hpi. Specifically, seven out of them were the cell wall depredating enzymes (e.g., polygalacturonase and endoglucanase 5), showing a specific upregulation in response to pathogen inoculation. Notably, one typical extracellular invertase (*PtCWI3*) (Potri.006G210600) was induced by Fs at 48 hpi and 72 hpi. Elevation of extracellular invertase expressions (activities) in plant–pathogen interactions is essential to modulate sugar partitioning for the pathogen colonization. CWI activity also triggers plant defense responses, including induction of PRs, metabolite-related gene expression, callose deposition, and cell death reduction [16]. The decreased genes observed in transcriptomic data mostly belong to xyloglucan endotransglucosylase/hydrolase protein (10), glucan endo-1, 3-β-glucosidase (5), β-galactosidase (5), and α-L-frucosidase 1(3) families at the same time point. Moreover, a homolog of UDP-glucuronic acid decarboxylase 2 (Potri.002G204400) and eight of 11 cellulose synthase-encoding genes were markedly downregulated in response to Fs infection. A shift in expression of two proline-rich glycoproteins, four LEA hydroxyproline-rich glycoproteins, and two syntaxin genes was also observed. Significantly, *Populus* roots response to Fs infection at 72 hpi was characterized by the hyperaccumulation of transcripts of cell wall biogenesis-related genes, including four pectate lyases, 11 PME enzymes, and 14 PMEIs that have been verified by qPCR (Figure 5c). A more recent report revealed that some genes involved in shikimate phenylpropanoid-lignin and cellulose biosynthesis pathways were characterized to consolidate cell wall structure in *Fusarium* infected plants [82]. Similarly, genes for cell wall structure have been reported to express during Fs infection in tomatoes [83]. These findings further reinforce our results.

Emerging evidence shows that the coordinated expression/activity of sucrose cleavage by CWIs and host hexose transporters act as essential indicators in plant–pathogen interactions [16]. Thus, pathogenic fungi use dynamic nutrient acquisition mechanisms to ensure a prolonged supply of carbohydrates and other nutrients through the regulation of sugar transporters from the host [84]. *Xanthomonas* bacteria secrete effector proteins to induce sugar efflux transporters’ expression, namely, sugars will eventually be exported transporters (SWEET) proteins led to sucrose accumulation in the apoplast of the infected rice [85]. In *Arabidopsis, AtSTP4/Atβfruct1* pairs are co-induced upon the biotrophic fungal infection [86]. In return, plant hosts can retrieve sugars from the infection niche by activating high-affinity sugar transporters. It has been reported that sucrose transporter-encoding genes were firstly downregulated in grapevine effected by *stolbur phytoplasma* to limit the sucrose release and then recovered by upregulation to provide necessary nutrients [87]. Specifically, we observed the strong upregulation of two of 5 SWEET bidirectional sugar transporters (Potri.001G060900 and Potri.015G101500) and a homolog of STP13 (Potri.010G089800) in response to the Fs infection at 72 hpi. These results substantiate sucrose’s active mobilization and homeostasis in Fs inoculated *Populus* roots, suggesting that ready-to-use sugars provided for pathogen growth and plant immunity likely depend on similar mechanisms in woody plants. Moreover, the biological roles of specific genes related to signaling, defense response, transcription regulation, and secondary and carbohydrate metabolism remain mostly unknown; the specificity of plant response derived from the well-studied pathological system might be exciting points to investigate in woody *Populus*.

## 5. Conclusions

The timing expression patterns of global genes in *Populus* roots infected with pathogenic Fs were systematically analyzed using transcriptomic sequencing. The markedly expressed genes in defined combinations were analyzed, and more active and drastic responses were identified upon Fs infection. The transcriptome’s molecular patterns represented a more robust activation of many well-known genes related to defense response, signaling transduction, TFs, and secondary and sugar metabolism. The highest number of responsive genes within the top significantly enriched pathways identified in *Populus* roots at the late stages of infection prompted necessity to inspect the altered metabolites and other bioactive compounds. To date, invertase inhibitors-like proteins involved in shaping the dynamics of apoplast-adapted pathogens remain unknown in woody *Populus*. In our study, the validated transcriptomic data may be useful to unravel the molecular basis of *Populus*–Fs interactions and develop new resistant mechanisms in *Populus* against Fs, including strategies that fine-tune sugar homeostasis and signaling through the post-transcriptional or post-translational control.

## Figures and Tables

**Figure 1 jof-07-00089-f001:**
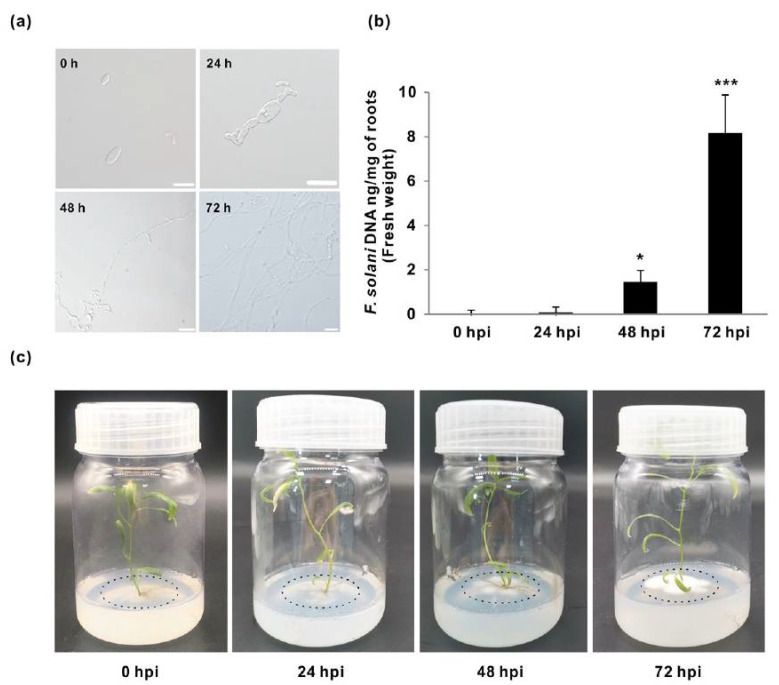
Growth phenotypes and the quantitative analyses of *Fusarium solani* (Fs) infection. (**a**) Examination of Fs conidial germination in the woody plant medium (WPM) during the time-coursed growth (0, 24, 48, and 72 h post-inoculation (hpi)) with scales of 0.01 mm (0 and 24 hpi) and 0.25 mm (48 and 72 hpi). (**b**) The quantitative PCR (qPCR) quantification of colonization efficiency of Fs inoculated to roots of the in vitro cultured *Populus*. (**c**) The growth patterns of *F. solani* at different time courses. Data represent mean values standard error (±SE) of three independent biological replicates. The asterisks indicate significant differences compared to the control (0 hpi) using Student’s *t*-test: *** *p* < 0.001 and * *p* < 0.05.

**Figure 2 jof-07-00089-f002:**
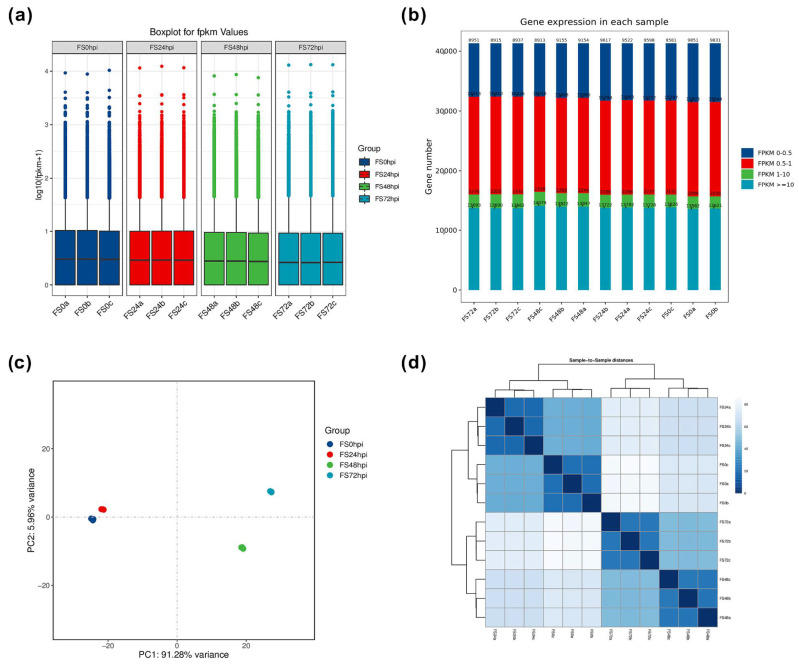
Statistical survey of the transcriptomic sequencing data sets under four different time courses of Fs infection. (**a**) The boxplot shows the composition of unique genes calculated by FPKM value. (**b**) The numbers and distributions of expressed genes with classified FPKM values. (**c**) The PCA shows the data variation between four different conditions. (**d**) Pearson’s tests show the data correlation within the same condition and different conditions. FPKM, the fragments per kb per million reads. PCA, principal component analysis.

**Figure 3 jof-07-00089-f003:**
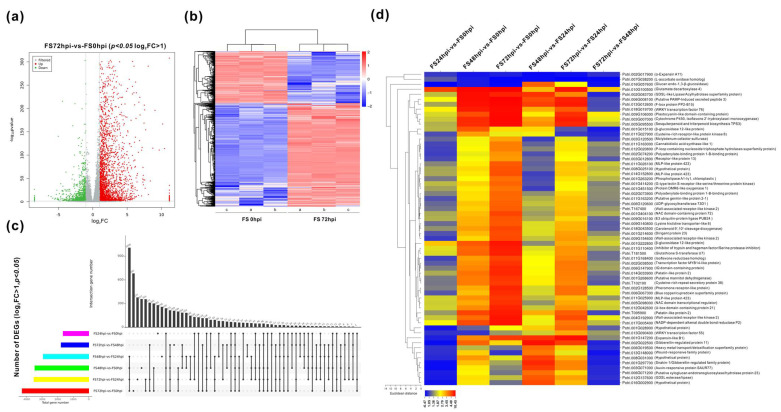
Dynamics numbers and changes of DEGs between different compared groups. (**a**) Numbers and distributions of DEGs with up-/downregulation up Fs infection at 72 hpi compared to the control (0 hpi). (**b**) Heatmap hierarchical clustering these DEGs. (**c**) Numbers of unique DEGs between the different comparison groups. (**d**) List of the significantly expressed genes and their annotations in six compared groups. The statistics of DEGs were based on the condition (log2FC > 1, *p* < 0.05).

**Figure 4 jof-07-00089-f004:**
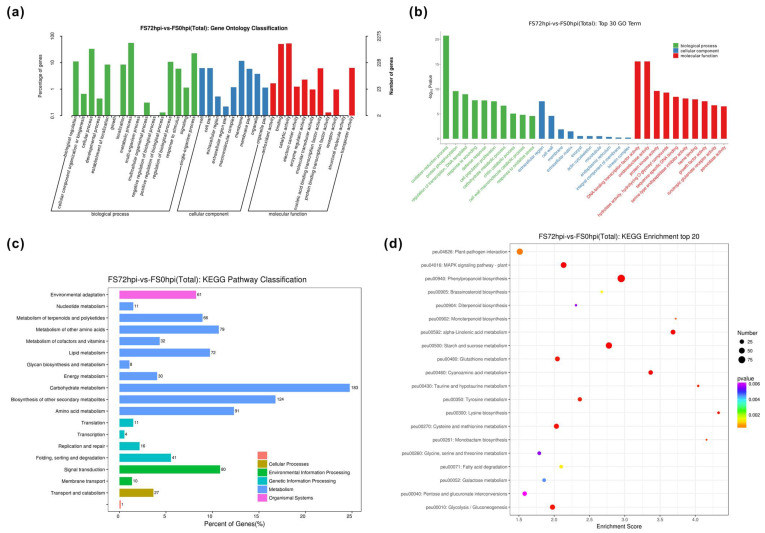
Statistical list of genes assigned to GO terms and KEGG pathways at 72 hpi. (**a**) DEGs are categorized into three major groups. (**b**) The composition of the top 30 GO terms. (**c**) Classification of KEGG pathways at 72 hpi. (**d**) The scatter plot shows detailed descriptions of the pathway within the top 20 KEGG enrichment. GO, gene ontology; KEGG, Kyoto Encyclopedia of Genes and Genomes. Analyses of the GO and KEGG enrichments were based on the condition (log2FC > 1, *FDR* < 0.05).

**Figure 5 jof-07-00089-f005:**
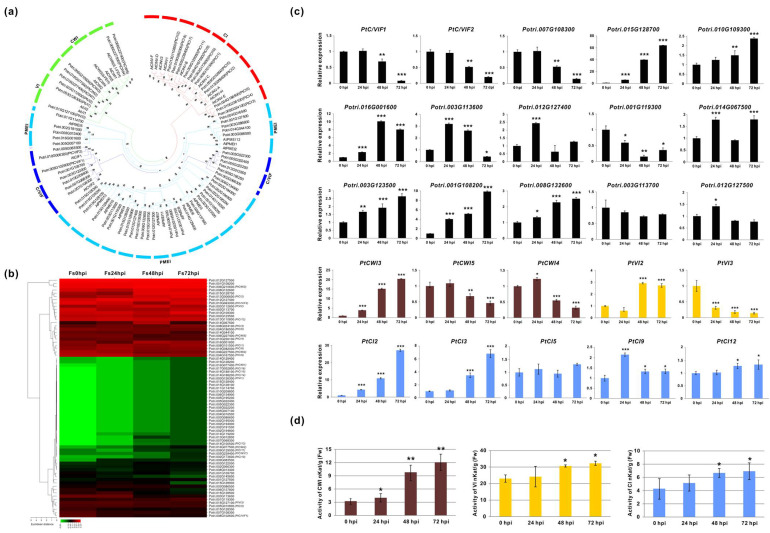
Phylogenetic relationships and quantification analyses of gene expression at four time courses of Fs infection. (**a**) Phylogenies of invertase and invertase inhibitor-like gene homologs between *Arabidopsis* and *Populus*. (**b**) The heatmap shows the hierarchical cluster of DEGs. (**c**) Validation of DEGs in Fs infected roots (24, 48, and 72 hpi) and the control (0 hpi) by qPCR. (**d**) Effects of invertase activities in roots upon Fs inoculation. The cell wall/vacuolar inhibitor of β-fructosidases (C/VIF), cell wall invertases (CWI), vacuolar invertases (VI), and cytosolic invertases (CI) are labeled in black, brown, yellow, and blue, respectively. The unrooted phylogenetic tree was constructed by MEGA7 using the neighbor-joining method based on 1000 bootstrap. Data represent mean values standard error (±SE) of at least three independent biological replicates. The asterisks indicate significant differences in comparison with the control using Student’s *t*-test: *** *p* < 0.001, ** *p* < 0.01, and * *p* < 0.05.

**Table 1 jof-07-00089-t001:** Output of sequencing statistics for samples in *Populus* roots infected with *F. solani*.

Sample	Raw Reads	Raw Bases	Clean Reads	Clean Bases	Valid Bases	Q30 ^1^	GC
FS0a	43.13M	6.47G	42.39M	6.00G	92.66%	93.13%	44.35%
FS0b	45.83M	6.88G	45.03M	6.38G	92.79%	92.98%	44.29%
FS0c	42.16M	6.32G	41.42M	5.86G	92.58%	93.10%	44.64%
FS24a	43.55M	6.53G	42.73M	5.99G	91.77%	92.73%	44.62%
FS24b	49.27M	7.39G	48.46M	6.86G	92.81%	93.10%	44.48%
FS24c	58.47M	8.77G	57.49M	8.15G	92.91%	93.19%	44.46%
FS48a	54.86M	8.23G	53.96M	7.67G	93.14%	93.11%	47.30%
FS48b	49.66M	7.45G	48.87M	6.94G	93.12%	93.37%	47.36%
FS48c	47.25M	7.09G	46.51M	6.59G	93.00%	93.45%	47.66%
FS72a	48.22M	7.23G	47.45M	6.71G	92.74%	93.33%	48.69%
FS72b	53.16M	7.97G	52.29M	7.40G	92.75%	93.35%	48.71%
FS72c	48.26M	7.24G	47.45M	6.71G	92.72%	93.20%	48.62%

^1^ The percentage of Qphred value > 30 within the raw bases. GC, the percentage of GC content in the whole sequence.

**Table 2 jof-07-00089-t002:** Statistical list of differentially expressed genes (DEGs) based on the condition of *p* < 0.05 and Log_2_FC^1^ > 1.

Groups	Up	Down	Total	Total (%)
FS24/FS0	979	680	1659	4.01
FS48/FS0	2669	810	3479	8.42
FS72/FS0	2870	1426	4296	10.39
FS48/FS24	2172	768	2940	7.11
FS72/FS24	2347	1193	3540	8.56
FS72/FS48	739	1048	1787	4.32

FC^1^, the fold change of normalized base mean value in compared two samples.

**Table 3 jof-07-00089-t003:** Numbers of DEGs and Gene Oncology (GO) terms at level 2 ^1^.

Groups	Total	Up	Down	Total	Up	Down
	Number of GO Term			Number of DEG		
FS24/FS0	61	59	47	888	565	323
FS48/FS0	79	76	62	1855	1447	408
FS72/FS0	90	76	65	2275	1501	774
FS48/FS24	101	98	79	1551	1105	446
FS72/FS24	91	86	75	1876	1184	692
FS72/FS48	57	50	51	923	362	561

^1^ The number of DEGs more than two annotated in GO term (false discovery rate (FDR) < 0.05).

**Table 4 jof-07-00089-t004:** List of the number of DEGs based on the KEGG enrichment at level 2 ^1^.

KEEG Term	ID	FS24/FS0 (All/Up/Down)	FS48/FS0 (All/Up/Down)	FS72/FS0 (All/Up/Down)
Phenylpropanoid biosynthesis	path:peu00940	27/16/11	69/59/10	92/75/17
Glycolysis/Gluconeogenesis	path:peu00010	4/3/1	34/30/4	36/33/3
Sugar metabolism ^2^	path:peu00520	8/4/4	36/34/2	29/24/5
α-Linolenic acid metabolism	path:peu00592	2/0/2	27/24/3	31/28/3
Cyanoamino acid metabolism	path:peu00460	11/6/5	23/17/6	26/21/5
Starch/sucrose metabolism	path:peu00500	20/13/7	48/35/13	61/40/21
Plant–pathogen interaction	path:peu04626	20/13/7	53/49/4	59/50/9
MAPK signaling pathway	path:peu04016	15/12/3	36/34/2	49/48/1
Hormone signal transduction	path:peu04075	19/12/7	50/0/50	51/40/11
Glutathione metabolism	path:peu00480	11/7/4	15/15/0	32/26/6
Pentose and glucuronate	path:peu00040	16/10/6	27/24/3	26/9/17
Galactose metabolism	path:peu00052	7/7/0	13/13/0	16/11/5
Cys and Met metabolism	path:peu00270	8/4/4	27/24/3	35/24/11
Gly, Ser, and Thr metabolism	path:peu00260	3/1/1	16/11/5	17/10/7
Tyr metabolism	path:peu00350	3/0/3	21/19/2	22/16/6
Carotenoid biosynthesis	path:peu00906	6/4/2	12/0/12	12/8/4
Zeatin biosynthesis	path:peu00908	11/5/6	11/7/4	11/5/6
Carbon fixation	path:peu00710	9/0/9	14/0/14	10/9/1
Brassinosteroid biosynthesis	path:peu00905	6/0/6	10/7/3	9/2/7
Monoterpenoid biosynthesis	path:peu00902	3/3/0	5/5/0	7/7/0
Lys biosynthesis	path:peu00300	1/1/0	6/6/0	9/6/3
Fatty acid degradation	path:peu00071	n.a	16/15/1	16/16/0
Diterpenoid biosynthesis	path:peu00904	5/3/2	9/0/9	8/5/3
Steroid biosynthesis	path:peu00100	4/3/1	5/2/3	4/0/4
Taurine and hypotaurine	path:peu00430	1/1/0	6/6/0	8/8/0

^1^ The number of DEGs annotated is more than two within the top KEGG term (*FDR* < 0.05). ^2^ Sugar metabolism of the amino and nucleotide; n.a, not applicable.

## Data Availability

The raw sequence reads of RNA-seq were deposited in the NCBI database with accession BioProject of PRJNA680933 and the accession BioSample, SAMN16927537, including twelve accession numbers of SRR13347970-981 for triplicate data of each Fusarium treatment (Fs0, Fs24, Fs48, and Fs72).

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
