# Peer review of "Transcriptomic Profiling of Populus Roots Challenged with Fusarium Reveals Differential Responsive Patterns of Invertase and Invertase Inhibitor-Like Families within Carbohydrate Metabolism"

_jof, 2021, doi:10.3390/jof7020089_

Round 1
Reviewer 1 Report
The authors have performed well-designed that can be improved with a little effort. My major issue with the paper has less to do with the question, why we do this experiment. Authors should make an effort to:
- Materials and Methods
Clarify the methodology, add statistical chapter.
- Discussion
Make an effort to concentrate / increase the discussion and make a solid discussion no chapters.

Author Response
The authors have performed well-designed that can be improved with a little effort. My major issue with the paper has less to do with the question, why we do this experiment. R1: Thanks for the comments. Several key issues addressed below help explain our research objective (highlighted in the re-uploaded manuscript). 1) Despite the relevance of forest species from the environmental, economic, and social point of view, our knowledge of the mechanisms underlying the forests growth and stress adaptation is still limited when compared to the crop plants (or Arabidopsis) (Tuskan et al., 2006). The infection of the necrotrophic fungal pathogen (e.g., Fusarium ) that causes root rot and vascular wilt lead to the substantial yield and quality loss in forest organisms has spurred a high demand and emergence for improvement of the host resistance to secure the tree biomass production. The recent advance in genomics and biotechnology provides new tools to unravel key defense regulatory processes in functional tree biology (Tsai et al., 2018; Plomion et al., 2016). Therefore, the improved understanding of underlying molecular mechanisms controlling entire plant-pathogen interactions is the primary strategy to manipulate the host resistance and reduce Fusarium pathogen's infection and damage in trees. 2) Populus has served as a model woody organism in perennial plants and forestry to research biology and molecular physiology, owning to high superiorities for plantation, biomass, and ecological functions (Jansson and Douglas, 2007). The defense response against Fs has been explored in some crops; however, little is known about the global atlas of root defense response in susceptible Populus, a model woody species. In a bid to rectify this situation, we conducted the RNA-sequencing and presented the transcriptome data set, which is one of the first steps to identify the altered gene expression in roots to understand potential defense mechanisms. We aimed to characterize highly DEGs during the host-pathogen interaction to reinforce the understanding of underlying defense mechanisms in woody Populus. (3) According to the processed transcriptomic data, the carbohydrate metabolism-related gene classification was mapped to be the largest functional category and the most predominant pathway. The gene expression of selected enzymes involved in fine-tuning regulation of the plant sugar metabolism confirmed is supposed to be critical players in the race for nutrients between the host plant and invading pathogens, informing us further to inspect the impact on the sugar metabolites and other bioactive compounds. Therefore, the timing expression patterns of global genes in Populus roots in response to pathogenic Fs provide novel insights into regulatory functions or signaling roles of clustered genes involved in biotic stress adaptation in woody model organisms or forest trees. Authors should make an effort to: Materials and Methods, Clarify the methodology, add a statistical chapter. R2: Thanks. A detailed description of methods and statistical analyses with references cited have been provided in the Materials and Methods for the transcriptomic sequencing. The standard statistics of significant variation (e.g., student t-test) have been clarified in qPCR and enzyme assay (Fig. 1 and Fig. 5). Please check the uploaded manuscript with tracked changes. Discussion Make an effort to concentrate / increase the discussion and make a solid discussion no chapters. R3: This is a useful comment. Transcriptomic research aims to provide a more extensive view on a systematic transcript response in plants host upon the fungal pathogen infection. The transcriptome's molecular patterns represented a more robust activation of many well-known genes related to defense response, signaling transduction, TFs, and secondary and sugar metabolism. We typically consider the RNA-seq data observed and attempt to put them into perspective with previously known literature in the field, prompting testable hypotheses of various gene functions in Populus. Here, a total of 923 Populus genes were identified to be highly differentially expressed in response to the Fs infection (FS72/FS48). Of these genes, approximately 839 were drastically enriched in 29 GO terms, and 240 were categorized into 21 KEGG pathways (Table S3). These findings allow us to place DEGs into the top five representative classifications (e.g., signaling transduction, defense responsive-related, TF regulation, secondary metabolism, and carbohydrate metabolism enriched pathways) that have been divided into three significant chapters for further discussion. Furthermore, as we have addressed in the discussion, the molecular mechanisms controlling entire infection and defense between plant hosts and microbial pathogen interactions are even more complicated extra- and intracellular processes than we envisioned. The host perceiving biotic signal to trigger defense, immune response, and resistance involved many genes/families within a diverse range of signaling pathways, phytohormone pathways, and primary/secondary metabolite pathways. We cannot exclude the crosstalk between various pathways and the pleiotropic effect in plant hosts. Besides, the fungal pathogenesis (e.g., enzymes, toxic molecules, effectors, and signals) mechanisms is another crucial issue that remains to be deciphered urgently. Thus, it could make no sense to primarily focus on the specific gene(s) in the discussion regime. As specific sucrose fine-tuning genes were highly regulated upon the fungal factors, explore their potential role by analyzing the possible phenotypes of mutants and molecular interaction with inhibitors, TFs, phytohormone under the biotic stress exposure will be the subject of future work.
Reviewer 2 Report
The authors in their manuscript entitled "Transcriptomic Profiling of Populus Roots Challenged with Fusarium Reveals Differential Responsive Patterns of Invertase and Invertase Inhibitor-like Families within Carbohydrate Metabolism” confirmed that plants, after biotic stress, induce several genes associated with primary metabolic pathways (e.g. synthesis and degradation of corboigrates) and the involvement of metabolic pathways in the stress response. Many studies have evaluated the response and pathological reactions to fungi in woody plants but relatively few have investigated molecular interactions. This is an interesting study and the authors provides new study perspectives and identifies new additional components involved in defense responses, as well as the detailed characterization of the mechanisms underlying these responses.
This is an excellent report, on very thorough research. The reviewed study is well designed and the data is reliable. Therefore, the content of the manuscript have value for the publication in J of Fungi.
Author Response
The authors in their manuscript entitled "Transcriptomic Profiling of Populus Roots Challenged with Fusarium Reveals Differential Responsive Patterns of Invertase and Invertase Inhibitor-like Families within Carbohydrate Metabolism” confirmed that plants, after biotic stress, induce several genes associated with primary metabolic pathways (e.g. synthesis and degradation of corboigrates) and the involvement of metabolic pathways in the stress response. Many studies have evaluated the response and pathological reactions to fungi in woody plants but relatively few have investigated molecular interactions. This is an interesting study and the authors provides new study perspectives and identifies new additional components involved in defense responses, as well as the detailed characterization of the mechanisms underlying these responses. This is an excellent report, on very thorough research. The reviewed study is well designed and the data is reliable. Therefore, the content of the manuscript have value for the publication in J of Fungi. R: Thanks for your comments.
Reviewer 3 Report
Comments to the author
The study is quite impressive and scientifically elaborated. I advise some minor corrections:
- Please add the references for the methods that you used in the material and methods section.
- Improve the quality of some figures.
- Ensure the uniformity in the units (e.g. mg/mL, µL) throughout the MS.
- Check the references in accordance with the journal style.
Author Response
Comments to the author The study is quite impressive and scientifically elaborated. I advise some minor corrections: Q1: Please add the references for the methods that you used in the material and methods section. R1: Thanks. A detailed description of methods, including cited references, has been inserted into the Materials and Methods. Please check the uploaded manuscript with tracked changes. Q2: Improve the quality of some figures. R2: Thanks. The resolution of all images in the manuscript has been improved to over 800 dpi. These figures may be visualized clearly by manually enlarging the size. Q3: Ensure the uniformity in the units (e.g. mg/mL, µL) throughout the MS. R3: Many thanks. These typos have been corrected, and please check the newly uploaded manuscript. Q4: Check the references in accordance with the journal style. R4: Thanks. Based on the criteria of JoF, each reference has been double-checked and formatted in the updated manuscript.
Round 2
Reviewer 1 Report
Dear Authors,
very good work. It is accept in this form